# Enhanced mosquito vectorial capacity underlies the Cape Verde Zika epidemic

**Noah H. Rose[1,2]ᴑ***, **Stéphanie Dabo[3]ᴑ, Silvânia da Veiga Leal[4], Massamba Sylla[5], Cheikh T. Diagne[6], Oumar Faye[6], Ousmane Faye[6], Amadou A. Sall[6], Carolyn S. McBride[1,2], Louis Lambrechts[3]***

1 Department of Ecology & Evolutionary Biology, Princeton University, Princeton, New Jersey, United States of America, 2 Princeton Neuroscience Institute, Princeton University, Princeton, New Jersey, United States of America, 3 Institut Pasteur, Université Paris Cité, CNRS UMR2000, Insect-Virus Interactions Unit, Paris, France, 4 Laboratório de Entomologia Médica, Instituto Nacional de Saúde Pública, Praia, Cabo Verde, 5 Department of Livestock Sciences and Techniques, University Sine Saloum El Hadji Ibrahima NIASS, Kaffrine, Senegal, 6 Institut Pasteur Dakar, Arbovirus and Viral Hemorrhagic Fevers Unit, Dakar, Senegal

ᴑ These authors contributed equally to this work.
* noahr@princeton.edu (NHR); louis.lambrechts@pasteur.fr (LL)

## Abstract

The explosive emergence of Zika virus (ZIKV) across the Pacific and Americas since 2007 was associated with hundreds of thousands of human cases and severe outcomes, including congenital microcephaly caused by ZIKV infection during pregnancy. Although ZIKV was first isolated in Uganda, Africa has so far been exempt from large-scale ZIKV epidemics, despite widespread susceptibility among African human populations. A possible explanation for this pattern is natural variation among populations of the primary vector of ZIKV, the mosquito *Aedes aegypti*. Globally invasive populations of *Ae. aegypti* outside of Africa are considered effective ZIKV vectors because they are human specialists with high intrinsic ZIKV susceptibility, whereas African populations of *Ae. aegypti* across the species' native range are predominantly generalists with low intrinsic ZIKV susceptibility, making them less likely to spread viruses in the human population. We test this idea by studying a notable exception to the patterns observed across most of Africa: Cape Verde experienced a large ZIKV outbreak in 2015 to 2016. We find that local *Ae. aegypti* in Cape Verde have substantial human-specialist ancestry, show a robust behavioral preference for human hosts, and exhibit increased susceptibility to ZIKV infection, consistent with a key role for variation among mosquito populations in ZIKV epidemiology. These findings suggest that similar human-specialist populations of *Ae. aegypti* in the nearby Sahel region of West Africa, which may be expanding in response to rapid urbanization, could serve as effective vectors for ZIKV in the future.

**Data Availability Statement:** The raw sequencing data are available in the Sequence Read Archive repository under accession number PRJNA882905. The raw experimental data plotted in Fig 1B–1C are provided in the S1 Data file.

Mosquito-borne arboviruses represent a major and growing threat to public health, causing sickness in over 100 million people every year [1]. *Aedes aegypti* is the primary vector of the viruses that cause dengue, chikungunya, Zika, and yellow fever across the global tropics, where it has aggressively invaded urban habitats over the last several hundred years [2,3]. The overall ability of mosquitoes to spread arboviruses is well described by the concept of vectorial

**Funding:** This work was supported by the European Union's Horizon 2020 research and innovation programme under ZikaPLAN grant agreement no. 734584 (to LL), Agence Nationale de la Recherche (grant ANR-18-CE35-0003-01 to LL), the French Government's Investissement d'Avenir program Laboratoire d'Excellence Integrative Biology of Emerging Infectious Diseases (grant ANR-10-LABX-62-IBEID to LL), and the US National Institutes of Health (grant NIDCD R00-DC012069 to CSM). NHR was supported by a Helen Hay Whitney Postdoctoral Fellowship. CSM is a New York Stem Cell Foundation – Robertson Investigator. The funders had no role in study design, data collection and interpretation, or the decision to submit the work for publication.

**Competing interests:** The authors have declared that no competing interests exist.

**Abbreviations:** CI, confidence interval; FFU, focus-forming unit; GLMM, generalized linear mixed model; SNP, single-nucleotide polymorphism; ZIKV, Zika virus.

capacity, which mathematically codifies the key entomological parameters driving transmission [4]. However, burdens of arboviral disease vary considerably across *Ae. aegypti*'s range, and the factors that lead to differences in the emergence and spread of arboviruses at both local and global scales are incompletely understood [5–7].

The explosive emergence of Zika virus (ZIKV) in the last 15 years represents an ideal example of the wide variation observed in arbovirus epidemiological dynamics across the global tropics [8]. ZIKV was first isolated from a sentinel monkey in the Zika forest, Uganda, in 1947 [9]. Retrospective serological surveys detected ZIKV circulation in the human population in Mali in the late 1990s [10] and in Gabon in 2007 [11], but no major disease outbreak was reported. The first recorded human epidemic of ZIKV occurred in Yap, Micronesia, in 2007, where *Aedes hensilli* is thought to have been the main vector [12]. In the following decade, ZIKV rapidly spread across the Pacific to the Americas, where it caused major epidemics vectored by *Ae. aegypti*, with a particularly severe outbreak in Brazil that culminated in 2016 [8]. These epidemics were associated in many cases with severe clinical outcomes, including Guillain–Barré syndrome and microcephaly in infants caused by ZIKV infection during pregnancy [13]. Across this same period, there has been no corresponding major ZIKV outbreak across Africa [8], despite presumably numerous virus introductions from the ongoing epidemic in the Americas. For example, introduction of ZIKV from Brazil resulted in autochthonous transmission and 4 confirmed cases in Angola in 2016 to 2017 [14,15]. A notable exception to this pattern occurred in Cape Verde, which experienced the first documented Zika epidemic in Africa during 2015 to 2016 following introduction of a ZIKV strain from the Americas [16]. One possible explanation for the lower incidence of Zika in Africa could be preexisting immunity in places where ZIKV has circulated for a longer time, but serological testing from across Africa has consistently found widespread susceptibility to ZIKV in people [17,18].

An alternative explanation for differences in ZIKV emergence and spread could come from variation among mosquito populations. *Ae. aegypti* originated in Africa, but then spread out of Africa and across the global tropics within the past 500 years [3,19]. Across its invasive range in the Americas and Asia, *Ae. aegypti* is a remarkably effective vector of arboviruses due to its strong specialization on human hosts and habitats [6,20,21]. This invasive human-specialist form, also known as the *Ae. aegypti aegypti (Aaa)* subspecies, breeds in artificial water containers and strongly prefers human host odor, allowing it to efficiently spread arboviruses from human to human [6,22,23]. However, throughout most of sub-Saharan Africa, populations of the ancestral subspecies *Aedes aegypti formosus (Aaf)* maintain a generalist ecology [24]. *Aaf* populations breed in both natural and human habitats and are attracted to a wide variety of vertebrate hosts [23–26]. Beyond its behavioral and ecological differences, *Aaf* is also less susceptible to ZIKV than *Aaa* [27]. Overall, the generalist ecology and low ZIKV susceptibility of *Aaf* may have contributed to hindering ZIKV emergence in Africa until now, despite the high epidemic potential of local ZIKV strains [28].

Although most populations of *Ae. aegypti* in Africa remain generalists with ancestral *Aaf* features [24], there is one region of Africa where human-specialist populations thrive: in the West African Sahel, south of the Sahara Desert. Here, there is little natural habitat for generalists across the long dry seasons. Instead, human-specialist populations of *Ae. aegypti* breed in human water storage containers and show both strong attraction to human hosts and high susceptibility to ZIKV infection [24,27]. Although these populations have a human-specialist ecology and likely descended from the original human-specialist population that spread out of Africa [24,29], they show a mixed genomic signature that reflects their lack of physical isolation from nearby generalists, with some degree of ancestry shared with human-specialist populations outside of Africa, as well as a strong contribution from nearby West African generalist populations [24]. Outside of the West African Sahel, a similar *Aaa*-like genomic signature can

be found in Angola and coastal Kenya; however, these populations have otherwise distinct ancestries and histories [24,30]. In densely populated urban areas across Africa, *Ae. aegypti* populations can also have substantial human-specialist ancestry and show increased attraction to human hosts relative to nearby rural populations [24].

Cape Verde is a group of small islands off the coast of West Africa, close to the contact zone between human-specialist *Ae. aegypti* populations in the West African Sahel and the more widespread African generalist populations. The genomic background and vector status of *Ae. aegypti* in Cape Verde has not been comprehensively characterized to date, but the 2015 to 2016 epidemic indicates that local mosquito populations are presumably effective ZIKV vectors [16]. There are no other known ZIKV vectors besides *Ae. aegypti* in Cape Verde; *Ae. caspius* is present but is not a known ZIKV vector [31,32]. *Ae. aegypti* was first observed in Cape Verde in 1931, and mitochondrial DNA analyses suggest that it may have been recently introduced from West Africa [32,33]. However, genome-wide patterns of ancestry and corresponding patterns of host preference and arbovirus susceptibility remain unknown. Here, we use genomic analysis, behavioral assays, and experimental infections to test the hypothesis that Cape Verdean *Ae. aegypti* mosquitoes are more effective ZIKV vectors than the *Aaf* populations that predominate across most of Africa. We find that a population of *Ae. aegypti* recently sampled in Praia, Cape Verde (CPV) shows a strong genome-wide signature of human-specialist ancestry, similar to that seen in nearby human-specialist populations in the West African Sahel and not in generalist populations found across most of Africa. We identify a correspondingly robust preference for human hosts and a higher ZIKV susceptibility than generalist counterparts.

We sequenced the genomes of 15 individual *Ae. aegypti* specimens from the CPV population to 20× average sequencing depth (detailed experimental procedures are described in S1 Text). These individuals showed the strongest genomic affinity with West African *Aaf* populations, as 64% of average genome-wide ancestry across individuals was attributable to this ancestry component (Fig 1A, dark blue). However, CPV also contained a substantial amount of *Aaa* ancestry (23%; Fig 1A, red). This signature suggests that these populations may be derived from, or share a common history with, nearby human-specialist populations in the West African Sahel, where similar levels of *Aaa* ancestry are associated with strong preference for humans and increased susceptibility to ZIKV [24,27]. For example, human-preferring populations of *Ae. aegypti* from Thiès and Ngoye, Senegal have 22% and 37% *Aaa* ancestry, respectively, across the same single-nucleotide polymorphism (SNP) panel. Interestingly, unlike nearby West African populations, CPV contained substantial East/Central African ancestry (14%; Fig 1A, light blue). Similar patterns of ancestry have been observed in Libreville, Gabon (Fig 1A), and Luanda, Angola [30].

We used live host 2-port olfactometer trials to assay host preference in CPV relative to previously characterized human-specialist (*Aaa*) and generalist (*Aaf*) populations. A CPV colony was initiated from field-collected eggs and used in all subsequent laboratory assays within 2 generations of colonization. In the behavioral assays, we placed female mosquitoes from different laboratory colonies into a 50 × 50 × 80 cm chamber, where they used host odor to choose between 2 exit holes, one leading to a human host, and one leading to a guinea pig. We used guinea pigs because of their good temperament and history of use in similar experiments that have reliably separated *Aaa* and *Aaf* [22,24,25]. However, previous work has shown that olfactometer trials using alternative hosts (e.g., chicken, quail, laboratory rat, African grass rat) can recover the same behavioral variation [22,24,25]. Similarly, although these trials were carried out with a 31-year-old European-American male (the same human host used for most trials in our earlier study; [24]), previous work indicates that variation in host preference across the *Aaa*–*Aaf* continuum are consistent in trials with human hosts of different ethnicities, sexes,

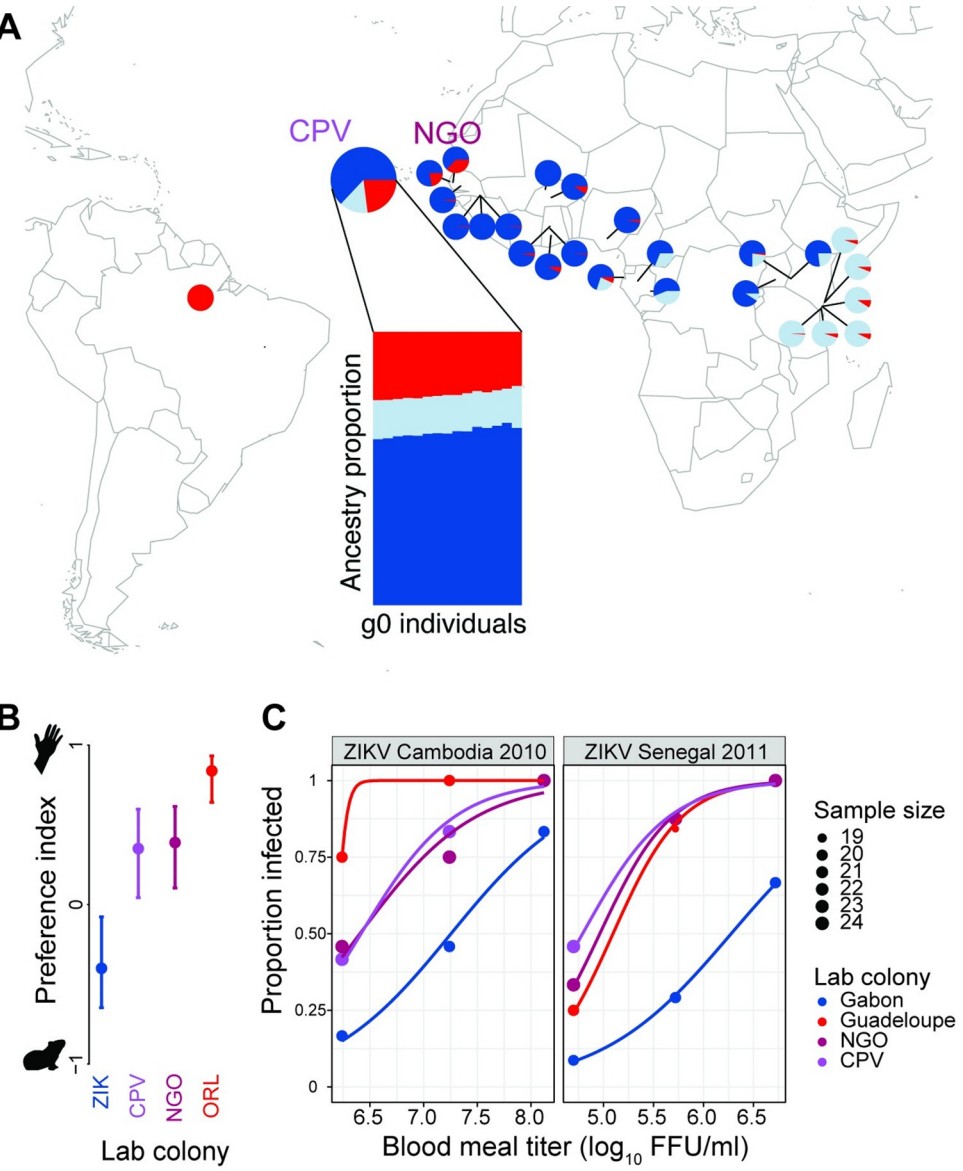

**Fig 1. Enhanced ZIKV vectorial capacity of *Ae. aegypti* in Cape Verde. (A)** In Cape Verde, *Ae. aegypti* has substantial human-specialist ancestry (23% on average, red). Pie charts show population averages for ADMIXTURE (K = 3) fractions of ancestry from human specialists (red), West African generalists (dark blue), and East/Central African generalists (light blue), in populations across *Ae. aegypti*'s native range in sub-Saharan Africa. Ancestry estimates for Cape Verde (CPV) are newly generated, whereas non-CPV estimates are reanalyzed from [24]. The human-specialist component dominates *Ae. aegypti* populations outside of Africa (e.g., Santarem, Brazil). Inset bar chart shows individual fractions of ancestry from the same 3 ancestry components in CPV, calculated across the same set of SNPs. CPV also shows ancestry from the East/Central African component (light blue), unlike nearby West African populations. The base layer of the map is from https://urldefense.com/v3/__https://cran.r-project.org/web/packages/maps/index.html__;!!JFdNOqOXpB6UZW0!sDKfbuINRVtMqxn5q2Db55SN93DxP6cPYdDUDB-Xy-GHDAIhO8GAhHf5FEJstypYbq4eF3PhXs9wPufm5y4eFCa5$. **(B)** In Cape Verde, *Ae. aegypti* shows a robust preference for human hosts. Preference indices are calculated from beta-binomial analysis of repeated 2-port live host olfactometer trials (4 trials of 87 to 102 females for each colony). CPV was tested alongside a generalist field-derived *Ae. aegypti* colony from Zika, Uganda (ZIK), a human-specialist colony from Ngoye, Senegal (NGO), and a human-specialist colony from Orlando, Florida (ORL). Error bars show 95% confidence intervals. **(C)** In Cape Verde, *Ae. aegypti* displays higher ZIKV susceptibility than a generalist colony, similar to human-specialist colonies. Dose–response curves represent the proportion of ZIKV-infected mosquitoes 7 days post-oral challenge as a function of the blood meal titers expressed in $\log_{10}$-transformed focus-forming units (FFU)/ml. Susceptibility was assessed for 2 ZIKV strains representing the African lineage (Senegal 2011, right) and the Asian lineage (Cambodia 2010, left). CPV was tested alongside a reference ZIKV-resistant colony from Gabon, a reference ZIKV-susceptible colony from Guadeloupe, and a colony from Ngoye, Senegal (NGO) with intermediate ZIKV susceptibility. Lines are logistic regression fits for each mosquito colony. The raw experimental data plotted in the figure are provided in S1 Data.

and ages [24,25]. Across olfactometer trials, we found strong evidence for variation among test colonies in preference for humans (GLMM likelihood-ratio test: $P$ = 0.001). CPV showed a robust preference for human odor (Fig 1B) and was significantly more likely to seek humans than the reference *Aaf* strain from Uganda (ZIK; Fig 1B, GLMM: $P$ = 0.0002), which preferred the guinea pig odor (Fig 1B). CPV showed similar levels of preference to Senegalese human specialists (NGO; Fig 1B) but was not as strongly attracted to humans as the human-specialist laboratory strain ORL, originally derived from Florida (Fig 1B, GLMM: $P$ = 0.0005).

We performed experimental infections to assess the ZIKV susceptibility of the CPV colony using previously characterized colonies from Gabon and Guadeloupe as references [27]. The *Aaf* colony from Gabon is poorly susceptible, whereas the *Aaa* colony from Guadeloupe and the NGO colony are highly susceptible. The CPV colony showed higher susceptibility to ZIKV strains from both the Asian lineage (Cambodia 2010 strain) and the African lineage (Senegal 2011 strain) relative to the *Aaf* colony from Gabon (Fig 1C). Using the less infectious Cambodia 2010 ZIKV strain, the CPV colony was significantly more susceptible than the colony from Gabon (CPV: 50% oral infectious dose [$OID_{50}$] = 6.42 $\log_{10}$ focus-forming units [FFU]/ml, 95% confidence interval [CI] = 5.97 to 6.71; Gabon: $OID_{50}$ = 7.25 $\log_{10}$ FFU/ml, 95% CI = 6.88 to 7.63), whereas it was similar in susceptibility to the NGO colony and the *Aaa* colony from Guadeloupe (NGO: $OID_{50}$ = 6.41 $\log_{10}$ FFU/ml, 95% CI = 5.80 to 6.74; Guadeloupe: $OID_{50}$ = 6.18 $\log_{10}$ FFU/ml, 95% CI = undetermined to 6.24) (Fig 1C). Using the more infectious Senegal 2011 ZIKV strain, the CPV colony was also significantly more susceptible than the colony from Gabon (CPV: $OID_{50}$ = 4.79 $\log_{10}$ FFU/ml, 95% CI = 4.29 to 5.09; Gabon: $OID_{50}$ = 6.28 $\log_{10}$ FFU/ml, 95% CI = 5.90 to 6.86) and similar in susceptibility to the NGO and Guadeloupe colonies (NGO: $OID_{50}$ = 4.96 $\log_{10}$ FFU/ml, 95% CI = 4.63 to 5.24; Guadeloupe: $OID_{50}$ = 5.09 $\log_{10}$ FFU/ml, 95% CI = 4.79 to 5.39) (Fig 1C). In all cases, human-specialist ancestry was associated with increased susceptibility to ZIKV, as expected. Our earlier study showed that higher ZIKV susceptibility (measured by the infection rate) was associated with enhanced potential to transmit ZIKV from a viremic host [27].

Overall, *Ae. aegypti* in Cape Verde appears to be an effective vector of ZIKV, showing substantial human-specialist ancestry, and, correspondingly, both robust preference for humans and increased intrinsic susceptibility to ZIKV relative to generalist *Aaf* populations that predominate in Africa. These observations may help explain why Cape Verde experienced a major ZIKV epidemic in 2015 to 2016, while other African nations did not experience similar outbreaks. Other factors such as human population density and mosquito abundance could also play an important role in driving differences in the spread of ZIKV; however, Praia is a relatively small city of about 150,000 inhabitants where the recorded mosquito abundances are not unusually high [32]. An important role for mosquito population variation is also consistent with ZIKV circulation in Angola in 2016 to 2017, where the *Ae. aegypti* population has a similar genomic signature [30]. Only 4 ZIKV cases were confirmed by direct virus detection, but they provided evidence of autochthonous transmission in Angola following ZIKV introduction from Brazil [14,15]. A key future research question is whether the elevated human-specialist ancestry of *Ae. aegypti* in Angola is associated with differences in behavior and ZIKV susceptibility, as observed in Cape Verde.

*Ae. aegypti* in Cape Verde had very similar levels of preference for humans and ZIKV susceptibility to human-specialist *Ae. aegypti* from Ngoye, Senegal. This raises the question of why corresponding ZIKV outbreaks were not observed in Senegal in 2015 to 2016, or perhaps in locations with human-specialist *Ae. aegypti* in coastal Kenya. It is possible that such outbreaks simply went unnoticed by public health surveillance systems due to misdiagnosis and/ or underreporting. A large variety of factors affect arbovirus transmission, including variation in human immunity, climate factors, levels of urbanization, and the success of control efforts

[34]. One important difference could be patterns of trade and travel related to historical relationships between nations. The 3 Portuguese-speaking nations of Brazil, Cape Verde, and Angola were closely linked during the transatlantic slave trade, which is thought to have driven the introduction of human-specialist populations of *Ae. aegypti* to the Americas [3]. In the present day, these nations are still closely linked through trade and migration [35], and ZIKV was likely introduced to Cape Verde from northeastern Brazil [16]. The *Aaa*-like ancestry that we observed in Cape Verdean *Ae. aegypti* showed evidence of contributions from both West Africa and the species' invasive range (S1 Fig); the latter signal could reflect gene flow from Brazil, although more comprehensive sampling of the Americas and the west coast of Africa would be necessarily to pinpoint the precise sources of the observed ancestry.

Further linking Cape Verde to arbovirus systems outside of West Africa, we observed substantial ancestry in Cape Verdean *Ae. aegypti* from the East/Central *Aaf* ancestry component that is nearly absent from most of West Africa, but common further south, including in Angola [30]. This finding suggests a more complex history for *Ae. aegypti* in Cape Verde than a simple introduction from nearby Senegalese populations. Instead, the *Ae. aegypti* population in Cape Verde may have experienced incoming gene flow from populations further south, possibly through trade with Angola. Improved genomic sampling from Angola and other parts of Southern and Central Africa could clarify the precise origins of this signal. Overall, our results suggest that the genetic makeup of vector populations may be shaped by historical relationships and patterns of human activity in addition to ecological factors and spatial proximity.

Further highlighting a role for human activities, ecological projections based on United Nations estimates of human population growth suggest that many urban populations of *Ae. aegypti* may shift towards a human-specialist ecology in the coming decades [24,36]. Presently, levels of *Aaa* ancestry in these cities are not as high as those observed in Cape Verde (e.g., Ouagadougou, Burkina Faso: 9%) [24]. However, as this proportion increases, such cities may be at increased risk of ZIKV outbreaks.

The mechanisms underlying variation in ZIKV susceptibility in *Ae. aegypti* are still unknown. Given the generally mild effects of arboviruses on *Ae. aegypti*'s fitness [37], these differences may not be adaptive on the part of the mosquito host—instead, they may reflect indirect consequences of other physiological or immune adaptations through pleiotropy or genetic linkage, genetic drift between mosquito lineages, and/or viral adaptation. A previous genetic mapping study detected a signal of association between the second *Ae. aegypti* chromosome and differences in ZIKV susceptibility, but the corresponding genomic region is large, making it difficult to identify causal genes [27]. Future efforts to identify the specific genes driving these differences could provide key tools for monitoring and managing the risk of ZIKV outbreaks in the future.

Taken together, our results are consistent with a prominent role for variation among vector populations in shaping the emergence and spread of arboviruses, especially in *Ae. aegypti*'s native range of Africa, where vector populations vary widely in ecology, behavior, and genomics [38]. This population-level variation is not static but is directly shaped by human-driven processes like rapid urbanization [24]. For this reason, comprehensive characterization of population differences in vector status, and proactive surveillance of changing mosquito populations will play an important role in managing and reducing burdens of mosquito-borne disease.

## Ethics statement

Wild *Ae. aegypti* eggs from Cape Verde were collected and exported with permission from the Ministry of Agriculture and Environment (authorization No. 988, dated 23 October 2020), in compliance with the Nagoya protocol. Mosquito colony maintenance at Princeton University

used direct blood feeding on human arms, which was determined to not meet the definition of human subjects research (Princeton University IRB Non-Human-Subjects Research Determination #6870). Experimental mosquito infections at Institut Pasteur used human blood samples to prepare artificial infectious blood meals. Healthy blood donor recruitment was organized by the local investigator assessment using medical history, laboratory results, and clinical examinations. Biological samples were supplied through the participation of healthy adult volunteers at the ICAReB biobanking platform (BB-0033-00062/ICAReB platform/Institut Pasteur, Paris/BBMRI AO203/[BIORESOURCE]) of the Institut Pasteur in the CoSImm-Gen and Diagmicoll protocols, which had been approved by the French Ethical Committee Ile-de-France I. The Diagmicoll protocol was declared to the French Research Ministry under reference 343 DC 2008–68 COL 1. All adult subjects provided written informed consent.

## Supporting information

**S1 Fig. ADMIXTURE analysis of Cape Verdean *Ae. aegypti* indicates a complex mosaic of ancestries.** We previously found that when specifying 6 ancestry components (K = 6), ADMIXTURE identified 2 distinct human-specialist ancestry components—a putatively ancestral African human-specialist component (marked in orange) and a globally invasive component corresponding to human-specialist lineages that left Africa and spread across the global tropics (marked in red) [24]. Human-specialist ancestry in Cape Verde includes substantial contributions from each of these components—this may reflect gene flow from both the invasive range (likely Brazil) and the west coast of Africa (likely Senegal and/or Angola). However, more extensive sampling from both the Americas and native range would be necessary to conclusively identify precise source populations. The base layer of the map is from https://urldefense.com/v3/__https://cran.r-project.org/web/packages/maps/index.html__;!! JFdNOqOXpB6UZW0!sDKfbulNRVtMqxn5q2Db55SN93DxP6cPYdDUDB-Xy-GHDAIhO8GAhHf5FEJstypYbq4eF3PhXs9wPufm5y4eFCa5$.
(TIF)

**S1 Text. Detailed description of experimental procedures.**
(DOCX)

**S1 Data. Raw experimental data plotted in Fig 1B and 1C.**
(XLSX)

## Acknowledgments

We thank Catherine Lallemand for assistance with mosquito rearing. We are grateful to Diego Ayala, Christophe Paupy, and Davy Jiolle for initially providing the mosquito colony from Gabon; Anubis Vega-Rúa for initially providing the mosquito colony from Guadeloupe; and John-Paul Mutebi for initially providing the mosquito colony from Zika, Uganda. We thank Maria da Luz Lima Mendonça for facilitating the mosquito sampling in Cape Verde. We are grateful to the volunteers and the ICAReB staff for the human blood supply.

## Author Contributions

**Conceptualization:** Noah H. Rose, Carolyn S. McBride, Louis Lambrechts.

**Formal analysis:** Noah H. Rose, Louis Lambrechts.

**Funding acquisition:** Carolyn S. McBride, Louis Lambrechts.

**Investigation:** Noah H. Rose, Stéphanie Dabo, Silvânia da Veiga Leal.

**Project administration:** Louis Lambrechts.

**Resources:** Massamba Sylla, Cheikh T. Diagne, Oumar Faye, Ousmane Faye, Amadou A. Sall.

**Supervision:** Carolyn S. McBride, Louis Lambrechts.

**Visualization:** Noah H. Rose.

**Writing – original draft:** Noah H. Rose.

**Writing – review & editing:** Carolyn S. McBride, Louis Lambrechts.

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
