## [Editor Report · Decision Letter 0]

25 Jul 2022

Dear Dr. Lambrechts, 

Thank you for submitting your manuscript entitled "Enhanced mosquito vectorial capacity underlying the Cape Verde Zika epidemic" for consideration as a Short Reports by PLOS Biology.

Your manuscript has now been evaluated by the PLOS Biology editorial staff, as well as by an academic editor with relevant expertise, and I am writing to let you know that we would like to send your submission out for external peer review.

Once your full submission is complete, your paper will undergo a series of checks in preparation for peer review. After your manuscript has passed the checks it will be sent out for review. To provide the metadata for your submission, please Login to Editorial Manager (https://www.editorialmanager.com/pbiology) within two working days, i.e. by Jul 27 2022 11:59PM.

Kind regards,

Paula

Senior Editor

PLOS Biology

---

## [Decision Letter · Decision Letter 1]

15 Sep 2022

Dear Dr. Lambrechts,

Thank you for your patience while your manuscript "Enhanced mosquito vectorial capacity underlying the Cape Verde Zika epidemic" went through peer-review at PLOS Biology. Your manuscript has now been evaluated by the PLOS Biology editors, an Academic Editor with relevant expertise, and by several independent reviewers.

In light of the reviews, which you will find at the end of this email, we are pleased to offer you the opportunity to address the comments from the reviewers in a revision that we anticipate should not take you very long. We will then assess your revised manuscript and your response to the reviewers' comments with our Academic Editor aiming to avoid further rounds of peer-review, although might need to consult with the reviewers, depending on the nature of the revisions.

You will see that all the reviewers agree that this is an interesting work and that your data supports your conclusions. Reviewer #3 comments that the title and title of figure 1 are inaccurate due to the use of vectorial capacity, however, both the Academic Editor and reviewer #1 have commented on this and agree that the vectorial capacity use in the manuscript is correct. Please address the rest of the reviewers' concerns.

Please also address the following editorial and policy requests:

1. DATA POLICY:

You may be aware of the PLOS Data Policy, which r**equires that all data be made available without restriction**: http://journals.plos.org/plosbiology/s/data-availability. For more information, please also see this editorial: http://dx.doi.org/10.1371/journal.pbio.1001797

A) Supplementary files (e.g., excel). Please ensure that all data files are uploaded as 'Supporting Information' and are invariably referred to (in the manuscript, figure legends, and the Description field when uploading your files) using the following format verbatim: S1 Data, S2 Data, etc. Multiple panels of a single or even several figures can be included as multiple sheets in one excel file that is saved using exactly the following convention: S1_Data.xlsx (using an underscore).

B) Deposition in a publicly available repository. Please also provide the accession code or a reviewer link so that we may view your data before publication.

Regardless of the method selected, please ensure that you provide the individual numerical values that underlie the summary data displayed in the following figure panels as they are essential for readers to assess your analysis and to reproduce it: Figure 1ABC.

**Please also ensure that figure legends in your manuscript include information on where the underlying data can be found, and ensure your supplemental data file/s has a legend.**

2. We suggest a minor modification of the title: "Enhanced mosquito vectorial capacity underlies the Cape Verde Zika epidemic".

**IMPORTANT - SUBMITTING YOUR REVISION**

*Resubmission Checklist*

*Published Peer Review*

Sincerely,

Paula

---

Senior Editor

PLOS Biology

REVIEWS:

Reviewer #1: Arbovirus.

Reviewer #2: Aedes genomics and evolution.

Reviewer #3: Vector biology, arbovirus.

Reviewer #1: see attached

Reviewer #2: This manuscript describes the higher vector competence of Aedes aegypti mosquitoes from Cape Verde to Zika viruses and their higher preference to blood feed on humans with respect to other African mosquitoes. This report appears to be a simple complement of a recent work authored by the same first author (Rose et al., 2020).

While I agree that preference for humans is a factor that can contribute to higher vector competence of Cape Verde mosquitoes, I do not agree on the (exclusive) emphasis put on it on this manuscript. Please see below main comments, which will hopefully help authors to provide a less skewed and more comprehensive hypothesis on the occurrence of the 2015-2016 Zika outbreak in Cape Verde.

Lines 57-60. This sentence is exaggerated and negates all the work done by scientists as Ross and McDonald. Factors that influence the emerge and spread of vector-borne disease, including arboviruses, have well been understood and codified into the vectorial capacity equation. These factors include vector density and extrinsic incubation period besides vector competence, which appears to be the focus of this manuscript. I would agree that factors that influence vector competence are still poorly characterised and that preference to feed on humans influences the blood-feeding rate, which is another parameter already codified into the standard equation of vectorial capacity produced by Ross and MacDonald.

Lines 85-99. Are Aedes species other than Ae. aegypti found in Cape Verde, mimicking a similar situation as in Yap?

Line 101-103 and 187-189. The Sahel region is quite extensive and ranges from Senegal, Mali, Burkina Faso, Nigeria, Cameroon, Central African Republic, Chad and Ethiopia. African populations of Ae. aegypti that were analysed and found to be human specialists by Rose et al. 2020 are limited to few populations in Senegal, to my knowledge. These populations are also cited in this ms (line 142). Ae. aegypti populations from Kenya and Angola have a completely different genetic ancestry than those from Senegal as described by several papers from the group of Jeffrey Powell, thus cannot be equated to human-specialist populations from Senegal.

Line 158-162: In this experiment, mosquitoes from Cape Verde showed stronger preference to seek humans than the Aaf strain from Uganda, as shown in Fig. 1B. However, in the supplementary methods Rose et al. described the use of 31 European-American male human individuals as host to perform the behavioural trials, while no African human individuals were included, not even from Cape Verde. Although the evaluation on humans preference from Africa and out of Africa mosquitoes is not strictly one of the main goals of this study, they did not mention whether there is any biological reason suggesting that the strong human preference observed in Fig. 1B will not change regardless of the human host's geographical origin and/or whether such preference will vary with African people.

Lines 185-187. I agree with the authors that preference for humans and high intrinsic susceptibility to Zika virus of mosquitoes from Cape Verde with respect to Aaf populations are important factors that could have contributed to the Cape Verde ZIKV epidemics of 2015-2016, but also the density of both hosts and vectors are important. Any information on abundance and distribution of vectors in Cape Verde and human density? Please discuss.

Line 202-208. Among the populations analysed by Rose et al. 2020, there is a Brazilian population and this hypothesis could be easily tested. In the ancestry plot shown in Fig. 1, how much is the ancestry linking Cape Verde with Brazil vs other Ae. aegypti populations tested in Rose et al. 2020?

Reviewer #3: This manuscript aims to answer the question, why have large-scale Zika virus outbreaks not occurred in Africa. The authors provide provocative evidence to suggest that the underlying genetics of Aedes aegypti in Africa may be playing an important role. Aedes aegypti originated in Africa where it still persists today as a forest dwelling form (Aedes aegypti formosus) and an urban form (Aedes aegypti aegypti). As Aedes aegypti aegypti spread through the equatorial tropics it has become a predominately human-biting mosquito. However, in most of Africa, Aedes aegypti populations contain significant formosus genetic signatures. Aedes aegypti formosus has more catholic feeding preferences and has been shown to have reduced vector competence to Zika virus compared to Aedes aegypti aegypti populations from the Americas (for example). Interestingly, Aedes aegypti from Cape Verde, Africa —which experienced a large-scale ZIKV outbreak in 2015-2016—have a substantial Aedes aegypti aegypti genetic ancestry. The authors show that mosquitoes derived from Cape Verde populations have a stronger preference for human host odors similar to other human-specialist populations of Aedes aegypti. In addition, they show that mosquitoes derived from Cape Verde populations have significantly higher Zika virus infection rates than mosquitoes with different genetic backgrounds from Africa. All of which could portend future risk of Zika virus outbreaks for the African continent due to urbanization and expansion of mosquitoes with a similar genetic background as those found in Cape Verde. This manuscript is written well and the results mostly support the conclusions. In addition, this serves as an interesting foundation for extensive follow-up work.

My primary concern is with the title of the manuscript and the title to figure 1. It is inaccurate to state that mosquitoes from Cape Verde have "enhanced vectorial capacity" for Zika virus. Vectorial capacity is the basic reproductive rate of a vector-borne pathogen. Vectorial capacity and vector competence cannot be used interchangeably. Vectorial capacity takes into account all of the environmental, behavioral, cellular, and biochemical factors that influence the association between a vector, the pathogen transmitted by the vector, and the vertebrate host to which the pathogen is transmitted. Here, the authors assessed host preference and mosquito vector competence—two components of overall vectorial capacity but these are not sufficient to quantitatively determine whether one population of mosquitoes has enhanced vectorial capacity compared to another population.

In addition, some text should be devoted to justify using infection rates as a proxy for vector competence and limitations relevant to the random and inefficient nature of mosquito transmission. Indeed, it has been previously shown that Aedes aegypti with poor competence but high population density have been capable of sustaining outbreaks of arboviral diseases like yellow fever virus. Also, infection data are only from a single timepoint, why was day 7 chosen?

---

## [Editor Report · Decision Letter 2]

3 Oct 2022

Dear Dr. Lambrechts,

Thank you for the submission of your revised Short Reports "Enhanced mosquito vectorial capacity underlies the Cape Verde Zika epidemic" for publication in PLOS Biology. On behalf of my colleagues and the Academic Editor, Nora Besansky, I am pleased to say that we can in principle accept your manuscript for publication, provided you address any remaining formatting and reporting issues. These will be detailed in an email you should receive within 2-3 business days from our colleagues in the journal operations team; no action is required from you until then. Please note that we will not be able to formally accept your manuscript and schedule it for publication until you have completed any requested changes.

PRESS

Sincerely, 

Paula 

---

Senior Editor

PLOS Biology
